# The role of lifestyle in mitigating cognitive decline in older adults with cardiometabolic multimorbidity (CMM): A protocol of systematic review and meta-analysis

Junyang Song◉*, Frances O'Brien, Jessica Eustace-Cook, Sharon O'Donnell*

School of Nursing & Midwifery, Trinity College Dublin, the University of Dublin, Dublin, Ireland

* songj4@tcd.ie (JS); sodonne@tcd.ie (SO)

## Abstract

### Background

Cardiometabolic multimorbidity (CMM), the coexistence of two or more cardiometabolic diseases (CMDs), is increasingly recognized as a key risk factor for cognitive decline and dementia in older adults. Healthy lifestyle behaviors may mitigate these effects, but this remains poorly understood. Given the rising prevalence of CMM worldwide and the urgent need for effective dementia prevention strategies, a systematic review and meta-analysis are warranted to investigate whether and to what extent a healthier lifestyle can mitigate this risk.

### Methods

A comprehensive search of Embase, Medline, PsycINFO, CINAHL, Web of Science, and Scopus will be conducted to identify longitudinal observational studies reporting incident adverse cognitive outcomes in older adults with CMM, while also exploring the influence of lifestyle factors on such outcomes. Grey literature will be included, and no language or date restrictions will be applied. Retrieved records will be managed in Covidence, with two independent reviewers conducting all review processes. Data will be extracted using a predesigned extraction form. The ROBINS-E tool will be used to assess the risk of bias, and the GRADE approach will evaluate the certainty of evidence. If sufficient homogeneous data are available, a meta-analysis will be conducted; otherwise, findings will be narratively synthesized.

### Discussion

This systematic review and meta-analysis will provide comprehensive insights into the extent to which a healthier lifestyle may mitigate cognitive decline associated with

**Data availability statement:** No datasets were generated or analysed during the current study. All relevant data from this study will be made available upon study completion.

**Funding:** JS received the Trinity College Dublin-China Scholarship Council PhD Scholarship (Grant Number 202406920021). The funders had no role in study design, data collection and analysis, decision to publish, or preparation of the manuscript.

**Competing interests:** The authors have declared that no competing interests exist.

CMM. The findings may inform future intervention designs to promote healthy aging and reduce the risk of cognitive impairment among individuals with CMM.

## Prospero registration

**Prospero registration ID:** CRD420251000046. Registered on 26th February 2025.

## Introduction

The prevalence of cardiometabolic disease (CMD), including hypertension, diabetes, dyslipidemia, stroke, and heart disease, is rising as the population ages [1]. CMD is a well-recognized risk factor for cognitive impairment. People with CMDs face an increased risk of developing cognitive decline, cognitive impairment with no dementia (CIND) [2], and ultimately dementia [3–5]. However, people living with CMD often face multiple conditions simultaneously rather than single diseases, commonly known as cardiometabolic multimorbidity (CMM), which is defined as the coexistence of two or more CMDs [6]. Research shows that the risk of cognitive decline and dementia rises as the number of coexisting CMDs increases, suggesting a dose-response relationship [7–9]. The risk of cognitive impairment increases by 11% for each additional CMD [10]. In older adults, those with CMM, such as clusters including hypertension, diabetes, coronary heart disease, and stroke, have nearly a two-fold risk of developing dementia compared to those without multimorbidity [11]. A study using the Swedish Twin Registry reported a hazard ratio (HR) of 2.10 (95% CI 1.73 to 2.57) for dementia associated with CMM in people aged 60 and older over an 18-year follow-up [7]. Similarly, a 12-year follow-up study of Swedish older adults aged 60 and older found a slightly lower HR of 1.86 (95% CI 1.17 to 2.97) [12]. While data from the Health and Retirement Study, with a mean follow-up of 14.6 years, showed a higher HR of 3.27 (95% CI 2.06 to 5.21) in American participants with a mean age of 59.4 years [13]. These findings underscore the importance of preventing the accumulation of CMDs and addressing modifiable risk factors to preserve cognitive health over time in older people with CMM.

The 2020 Lancet Commission report estimated that up to 40% of dementia cases could be prevented or delayed by targeting modifiable risk factors [14]. Many studies have investigated cognitive health using composite cardiovascular health scores, including indicators of both CMDs and lifestyle factors such as exercise, diet, and sleep [15,16]. While composite scores are useful for predicting cognitive outcomes in the general population, they may offer limited insight for tailoring interventions to individuals with CMM. In contrast, an emerging body of research has focused on the role of lifestyles in mitigating the effects of CMM on cognitive decline. Healthy lifestyles play key roles in preventing cognitive decline and delaying the onset of dementia, particularly among individuals younger than 60 years old [17]. Such findings are especially valuable for informing future preventive strategies for those already living with CMM. Existing research shows that maintaining healthy lifestyles, such as physical activity, limiting alcohol consumption, and avoiding smoking, can mitigate

the adverse effects of diabetes on brain ageing among people with poor cardiometabolic health [18]. Recent studies have coincidentally identified physical inactivity and excessive alcohol use as key accelerators of cognitive decline in individuals with CMDs [8]. Other research shows that an active lifestyle, including leisure activities and a rich social network, can delay dementia onset and reduce dementia risk by about 67% in older adults with CMDs [9]. A large multi-cohort study reported that cognitive decline worsened with both an increasing number of CMDs and a higher count of unhealthy lifestyles [8]. However, this study was limited to only three CMDs (i.e., diabetes, heart disease, and stroke) and three unhealthy lifestyle factors (i.e., physical inactivity, smoking, and excessive alcohol use), likely due to data constraints. Despite extensive literature exploring the role of lifestyle in cognitive decline within the context of cardiometabolic health, the evidence appears fragmented and inconsistent.

To date, no systematic review has comprehensively synthesized the evidence nor quantified the extent to which a healthier lifestyle mitigates CMM-related cognitive decline. A systematic review is therefore warranted to integrate existing findings. Variations in study design, including definitions of CMM, measurement of lifestyle factors, follow-up durations, and study settings, further underscore the need for a structured evaluation via systematic review. The objective of this review is to investigate whether a healthier lifestyle can mitigate the detrimental effects of CMM on cognitive health, including cognitive decline, CIND, and dementia. Where data permits, we will also explore characteristics that may influence the lifestyle's mitigating role, such as age group, CMM subtype, specific lifestyle factors, and follow-up duration. Findings from such a synthesis could provide an evidence-based foundation for developing targeted preventive strategies and tailored interventions for older individuals with CMM.

### Research questions

Does a healthier lifestyle mitigate the detrimental effects of CMM on cognitive health?.

## Method

### Protocol registration

This review protocol adheres to the Preferred Reporting Items for Systematic Reviews and Meta-Analyses for Protocols (PRISMA-P) guidelines (**S1 File**) [19]. The review will be conducted and reported according to the PRISMA 2020 checklist [20] and Meta-analyses of Observational Studies in Epidemiology (MOOSE) guidelines [21,22]. The study protocol has been registered on the International Prospective Register of Systematic Reviews (PROSPERO) on 26th February 2025 (registration number CRD420251000046).

### Search strategy

The search strategy for this review will encompass both published and unpublished studies. Six key databases: Embase, Medline (EBSCO), PsycINFO (EBSCO), CINAHL (EBSCO), Web of Science, and Scopus will be searched from inception up to March 2025. Additional relevant publications will be identified through a forward and backward search of key references. Grey literature, including unpublished studies, will also be searched using sources such as Google Scholar, ProQuest, and medRxiv. Studies published in all languages will be included to ensure methodological rigor, with Google Translate employed when necessary as a viable and accurate tool for translating in systematic reviews [23,24].

The index terms and keywords from the titles and abstracts will be used to develop a comprehensive search strategy, which will be validated by an experienced librarian. Key index terms examples include: (MH "Cardiometabolic Risk Factors") for CMM, (MM "Life Style+") for lifestyle factors, and ((MH "Dementia+") AND (MH "Prospective Studies+")) for cognitive decline in longitudinal studies. Detailed search strategies for all databases can be found in **S2 File**.

## Eligibility criteria

The study selection criteria based on the Population, Exposure, Comparator, and Outcomes (PECOS) standard will be employed to identify the eligible studies [25]. The focus will be on non-clinical contexts, such as community-based settings and nursing homes, to assess the influence of daily life lifestyles.

(1) Population: Older adults aged 50 years or above (at baseline) with CMM, defined as the coexistence of ≥ 2 CMDs [26]. We set the age cut-off because adults aged 50 and over experience a high prevalence of multimorbidity, with over 80% affected. [27], making this a critical window to examine the impact of CMD accumulation on cognitive health. Furthermore, as this population may not yet exhibit clinically significant cognitive impairment [28], studying the mitigating role of lifestyle factors is particularly relevant for informing future preventive strategies. Studies of individuals with dementia or Parkinson's disease at baseline will be excluded.

(2) Exposure: healthier lifestyle behaviors, including but not limited to diet, physical activity, sleep, smoking, drinking, social activities, technology use, cognitive activities, or a combination of these lifestyle factors.

(3) Comparator: Individuals with less healthy lifestyles within the context of CMM.

(4) Outcomes: Cognitive outcomes, including cognitive decline measured by changes in global or domain-specific cognitive scores (e.g., memory, executive function), incident cognitive impairment, or dementia (all cause or subtypes). Additional potential outcomes include cognitive risk biomarkers (e.g., APOE4, neuroimaging markers, and blood biomarkers such as tau and amyloid).

(5) Study design: Only studies using longitudinal observational design, such as prospective cohort studies and nested case-control studies, will be included in this review.

Studies will be excluded based on several criteria. Firstly, study designs such as trials, cross-sectional studies, case reports, and purely qualitative research will be excluded. Additionally, studies that are not published as full reports, such as study protocols and editorials, will not be considered. Lastly, research focusing on specific cognitive impairments that are unrelated to dementia or mild cognitive impairment (MCI), such as intellectual disabilities, will also be excluded.

## Studies selection

Following the search, all identified citations will be imported into Covidence systematic review software (https://www.covidence.org/), and duplicates will be removed automatically. At least two reviewers will independently screen the titles and abstracts according to the predefined inclusion and exclusion criteria. The full texts of potentially eligible studies will be retrieved for further evaluation. Two reviewers will perform full-text screening independently, with exclusion reasons documented. Discrepancies will be resolved through discussion and adjudicated by a third reviewer if necessary. The study selection process and reasons for exclusion will be recorded and presented in a PRISMA flow diagram (Fig 1).

## Data extraction

Two reviewers will independently extract data from the included studies using a standardized and pre-designed data extraction form to ensure consistency. The form will include study characteristics (first author, publication year, country, cohort name if has, study design), participant details (sample size, age, sex, baseline cognitive status), exposure details (subtypes of CMM, lifestyle factors), outcome characteristics (cognitive function changes measured via scales or neurocognitive tests or incidence of cognitive impairment), timing of assessments (baseline, follow-up duration), statistical data (mean and standard deviation (SD), risk estimates and their 95% confidence intervals (CIs), statistical methods, covariates, and missing data), and key findings. Extracted data will be recorded electronically in a secure spreadsheet (e.g., Excel) to facilitate easy access and analysis. The reviewers will work independently to minimize bias, with any

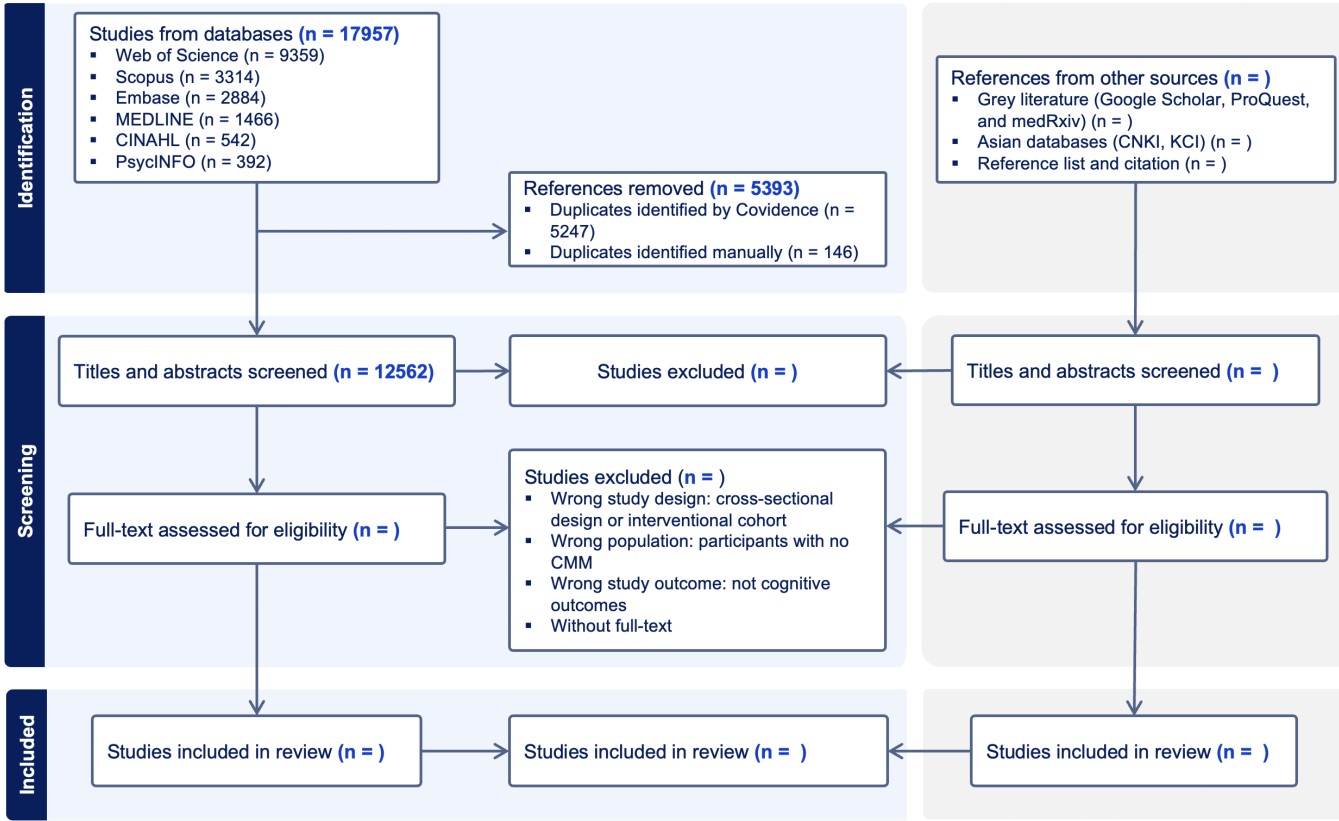

**Fig 1. PRISMA flow diagram.**

disagreements resolved by discussion. A third reviewer may be involved if disagreements persist. For studies lacking sufficient information, corresponding authors will be contacted via email at least twice to request additional details. The data extraction is expected to be completed before October 2025.

## Assessment of methodological quality

Two reviewers will independently conduct the risk of bias (RoB) assessments. Should any disputes emerge, they will be settled through discussion or consultation with a third reviewer. The results of the RoB assessments will be documented and reported in the systematic review, including a summary of the overall risk of bias for each included study, both narratively and in tabular format. The Risk Of Bias In Non-randomized Studies of Exposures (ROBINS-E) tool will be used [29], including seven domains of bias: (1) confounding, (2) measurement of the exposure, (3) selection of participants, (4) post-exposure interventions, (5) missing data, (6) measurement of the outcome, and (7) selective reporting of the results. The overall RoB is categorized as low, some concerns, and high, which will be visualized using the robvis tool [30].

## Data synthesis and statistical analysis

If sufficient homogeneous data are available, a meta-analysis will be conducted; otherwise, findings will be synthesized narratively and displayed in tables and figures to facilitate interpretation.

Hazard ratios (HRs) and their corresponding 95% CIs will be used as the unified effect estimates to examine the association between CMM and cognitive decline. Relative risks (RRs) and odds ratios (ORs) reported in the original studies

will be treated as equivalent to HRs, given the relatively low incidence of dementia [31]. Analyses will be performed on the log scale and pooled using random-effects models with restricted maximum likelihood (REML) [32]. To obtain more reliable uncertainty estimates under conditions of limited studies and substantial heterogeneity, we will apply the Hartung-Knapp-Sidik-Jonkman (HKSJ) method, which adjusts the standard error of the pooled effect and uses a t-distribution [33]. Where possible, dose-response meta-analysis will model continuous lifestyle measures. Heterogeneity will be reported with HKSJ-adjusted CIs, alongside the I² statistic and τ² estimates. In addition to the pooled estimates, we will report 95% prediction intervals to reflect the expected range of effects in future studies, offering greater practical interpretability [34]. Where sufficient data are available, meta-regression and subgroup analysis will be conducted to explore sources of heterogeneity and other potential effect modifiers, including sex, age group, sample size, CMM subtypes, lifestyle measurement, follow-up duration, and outcome definition.

Several sensitivity analyses will be conducted to ensure the robustness of the main results. First, we will apply the leave-one-out method, where each study is sequentially removed, and the meta-analysis is re-run to assess the effect of individual studies. Second, we will exclude studies that report ORs or RRs as effect estimates to maintain consistency in outcome measures. Third, we will exclude studies with follow-up durations of less than five years to avoid the potential risk of reverse causation [35]. Fourth, we will exclude studies with a high risk of bias to examine the robustness of the pooled results.

Publication bias in meta-analytic models will be assessed visually using funnel plots and quantitatively through a mixed-effects version of Egger's test for asymmetry, aligned with the principle of triangulation [36]. If publication bias is detected, the trim-and-fill method will be used to assess the impact of potentially missing studies on the pooled effect estimate [36,37].

All statistical analyses will be performed on RStudio version 4.4.1 (https://www.r-project.org/), using packages such as 'metafor' [38], 'mixmeta' [39]. A two-tailed p < 0.05 will be considered statistically significant.

## Certainty of evidence assessment

In a systematic review, certainty represents the confidence in the correct estimates of the effect. We will follow the Grading of Recommendations, Assessment, Development and Evaluation (GRADE) approach for grading the certainty of evidence [40,41]. It includes five domains for rating down certainty (i.e., risk of bias, inconsistency, indirectness, imprecision, and publication bias).

## Ethics and dissemination

This study involves analysis of previously published studies or fully anonymized data. Thus, ethical approval from the institutional review board is not required. The results may be disseminated through peer-reviewed publications and presentations at national and international conferences.

## Discussion

A systematic review and meta-analysis of observational studies is preferable to randomized controlled trials (RCTs) as they reflect more real-life settings [42]. For the aim of assessing the effect of CMM on cognitive outcomes and lifestyle mitigators, observational data can offer distinct advantages over RCTs, particularly in terms of the generalizability of findings and the ability to assess long-term outcomes, such as dementia. However, several potential challenges may arise and limit this review. Firstly, heterogeneity in the operational definitions of CMM, measures of cognitive function, and study designs could complicate the analysis. However, heterogeneity helps interpret evidence and identify meaningful variability when analyzed appropriately. Secondly, most studies report aggregate data, and access to individual participant data (IPD) is often restricted, limiting the ability to perform IPD meta-analyses for more flexible modelling of covariates and subgroup analyses [43]. Lastly, inconsistent reporting of lifestyle factors (e.g., exercise, diet, social activities) across studies

may hinder examining their mitigating role in the CMM-cognition relationship. Despite these challenges, a well-conducted systematic review and meta-analysis could inform the need for a more consistent methodology (i.e., better study designs, data collection methods, and analytical techniques), offering valuable insights to guide future research and thus improve the consistency and comparability.

The findings from this review will provide a clearer understanding of the relationships between CMM, cognitive health, and lifestyle factors. Supported by a pooled effect estimate through meta-analysis, it will present the overall effects of CMM on cognitive decline, thus enhancing external validity. Furthermore, future preventive strategies targeting lifestyle modifications may offer dual benefits, improving cardiometabolic health while reducing the risk of cognitive impairment, as suggested by prior scholars.

## Supporting information

**S1 File. PRISMA-P checklist.**
(DOCX)

**S2 File. Search strategy.**
(DOCX)

**S3 File. R code with simulated dataset.**
(DOCX)

## Acknowledgments

During the preparation of this work, the authors utilized AI-based tools to refine the language, enhancing readability and clarity. The final manuscript has been thoroughly reviewed and edited by the authors to ensure its accuracy and integrity. The authors take full responsibility for the content of the publication.

## Author contributions

**Conceptualization:** Junyang Song.

**Formal analysis:** Junyang Song.

**Funding acquisition:** Junyang Song.

**Methodology:** Junyang Song, Frances O'Brien, Jessica Eustace-Cook, Sharon O'Donnell.

**Supervision:** Frances O'Brien, Sharon O'Donnell.

**Writing – original draft:** Junyang Song.

**Writing – review & editing:** Frances O'Brien, Sharon O'Donnell.

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
