## [Decision Letter · Decision Letter 0]

18 Aug 2025

Dear Dr. SONG,

Thank you for submitting your manuscript to PLOS ONE. After careful consideration, we feel that it has merit but does not fully meet PLOS ONE’s publication criteria as it currently stands. Therefore, we invite you to submit a revised version of the manuscript that addresses the points raised during the review process.

We look forward to receiving your revised manuscript.

Kind regards,

Tai Dinh

Academic Editor

PLOS ONE

Journal Requirements:

[This work was supported by the Trinity College Dublin-China Scholarship Council PhD Scholarship (Grant Number 202406920021). The funders had no role in study design, data collection and analysis, decision to publish, or preparation of the manuscript.].

3. Thank you for stating the following in your manuscript:

[This work was supported by the Trinity College Dublin-China Scholarship Council PhD Scholarship (Grant Number 202406920021). The funders had no role in study design, data collection and analysis, decision to publish, or preparation of the manuscript.]

[JS received the Trinity College Dublin-China Scholarship Council PhD Scholarship (Grant Number 202406920021). The funders had no role in study design, data collection and analysis, decision to publish, or preparation of the manuscript.]

Additional Editor Comments:

Reviewer #1

Song and colleagues are planning to perform a systematic review and meta-analysis, here outlining their planned work in an overall well-written and constructed protocol. A PROSPERO version of the protocol is available, and as of 10th of July, it appears as if the work is at the stage of screening. Preferrably, the authors should have waited for at least the initial reviewer feedback before commencing the formal search and screening, but it is, at the same time, fully understandable that they have now, and in practical terms, this is often the reality. The topic is clinically relevant, with a novel angle, and I applaud the authors for submitting a detailed protocol! In general, I see no major issues with this protocol or the methodological approaches outlined; however, there are bits here and there that can benefit from some polishing and revision. I've outlined those below. Please apologize the many points, some of them are relatively trivial and have more the nature of questions.

PROSPERO (just three trivial questions)

- why did you switch the author order (between the PROSPERO protocol and this protocol)?

is any of the authors a methodologist (in meta-research/evidence synthesis)/librarian, or has a person with such competence been consulted throughout the planning of this SR/MA? (e.g., for double-checking the search queries or the like)

- is the "Current review stage" indeed up-to-date? Because as I see, the first three steps have been marked as "started", but neither of the first two have been marked as "completed"...

Title

- Why is the focus/(choice of word) on *mitigating* life-style factors? Usually, from what i see, life-style factors investigated are rather *risk* factors...

Abstract

- From the Background, it's not clear why a SR/MA is needed. Is it because the results from primary studies are conflicting? Is it that it's unclear which *specific* cognitive outcomes the associations are actually robust/consistent in? Something else?

- In Methods, you write "[...] in older adults with CMM"... but what's the control group then? Please clarify/specify this part

- In Methods, you write about two independent reviewers doing screening, but what about data extraction and quality assessment? Those should be performed in pairs as well

Introduction

- Please clarify the difference between cognitive decline and cognitive impairment

- I'd simplify line 61 to "two-fold"/"twice"

- The sentence on lines 66-68 feels unfinished..

- For the part on lines 88-93, please add country and mean age at baseline where it's missing

- On line 97, is it meant "[...] between CMM and cognitive outcomes [...]"?

- Wouldn't the sentence on lines 100-102 be better suited in the very beginning of that paragraph?

- Technically speaking (regarding lines 103-104), Alzheimer's disease is also a type of dementia.. So maybe just "dementia"? why limit yourselves in that regard?

- Again, on line 104, "[...] among older individuals with CMM [...]"... then who is the control group?

- Finally, on line 104, "[...] and 2) Identify [...]"... i think what you mean is to investigate? Also, again, I understand completely the interest in healthy lifestyle factors to adopt, but what about modifiable *risk* factors that can be avoided? it sounds mostly comprehensive when reading your text, but at times quite limiting (what you mean)...

Methods

- Which translation tools will be employed, and why?

- How will you deal with Google Scholar, given that it may easily return enormous amounts of records? Usually, SR/MA take the first 200-300 records or so; how is your plan, and why?

- Including gray literature is very good! but i wonder, is OpenGrey even searchable/indexed? It was more or less de facto defunct last time i checked... How about ProQuest Dissertations and Theses, OAIster, etc...

- I'd strongly recommend searching PubMed instead of Medline, as the former contains ALL of Medline-indexed records PLUS some preprints and other material that could potentially be relevant. Although Medline admittedly offers richer search syntax (e.g., NEAR/x), I'd argue that comprehensiveness outweighs in this case

- Any plans about ensuring non-western inclusion? E.g., i see you plan to use Web of Science, which is great. Please consider also searching KCI and SciELO through WoS (as i recall it doesn't search those by default, just the "core collection"). Also, please consider WHO Global Index Medicus, CABI Direct...

- I'm missing a search query for Google Scholar/OpenGrey

- Why the cut-off of ≥50 years (at baseline, i assume)?

- Hmm, reading the Comparators section, it remains a bit unclear to me, what will *actually* be investigated.. because based on lines 147-149, you are not searching for studies with control groups being without (as many) cardiometabolic diseases... so are you indeed *only* interested in populations where ≥2 CMDs are present?

- Are you sure that case-control studies qualify as longitudinal? Judging by the ROBINS-E paper, it's also not exactly geared for it ("Future work will produce variants of the tool for other epidemiologic study designs (e.g. case-control studies)."; https://pmc.ncbi.nlm.nih.gov/articles/PMC11098530). To strenghten the rigor and clinical utility of this SR/MA, i'd refrain from including case-control studies and keep it at prospective cohort studies, historical cohort data, case-cohort studies, nested case-control studies (i.e., within a cohort), etc..

- Please reconsider excluding letters to the editor. At least from my experience, the trend in some high-impact journals recently is to downgrade many good-quality articles to letters. Some of the authors then opt for putting most of the now "extra" material/data to suppl. material and online repositories (thus, the same data and rigor/detail is actually there, as in a "full-length" article)...

- Will you employ any definition of minimum follow-up length? Would it make sense for some outcomes, given how slowly they develop? Or is this something that you'll simply just comment on, if you see fit?

- I'd place (and also *do*) Data Extraction prior to Assessment of methodological quality

- I assume you might want to revise the estimated deadline on line 197?

- I agree that HR ≈ RR in case the outcome is rare (<15% or so), but i would say it's not quite the case otherwise (please see Table 2 in https://doi.org/10.7326/M16-2607)

- Please clarify (or rather, specify) the meta-analysis models you want to employ. I assume random-effects, although you specifically mention "mixed" modelling multiple times; what's the rationale and what setup are you expecting? Which method of assessment of between-study variance etc? Also, please consider looking into tau(2), as it's more robust than I2 for small sample sizes, if you fear such scenarios in particular

- Why meta-regression for RQ2? Do you really expect such data?

- Subgroup/sensitivity analyses seem fine, but why no sensitivity analysis excluding low-quality studies (for example)?

- Please consider making your R code publicly available, together with collected data (for meta-analysis); a great example/inspiration can be found here: https://journals.plos.org/plosmedicine/article?id=10.1371/journal.pmed.1003731

Discussion

- Is there a reference or a further specified rationale for what is said about IPD MA on lines 256-258? My personal point of view is that the heterogeneity seen in clinical context is so great that MAs in and of themselves are by default quite "rough" and always need to be examined cautiously, and i haven't seen proof of IPD MA making things better per se (in this context)

Reviewer #2

This is a well structured manuscript from abstract to reference. But, the author did not include any tables or diagram to show what was done to back up the analysis. No evidence of the research carried out was involved in this manuscript.

Reviewers' comments:

Reviewer's Responses to Questions

**Comments to the Author**

1. Does the manuscript provide a valid rationale for the proposed study, with clearly identified and justified research questions?

Reviewer #1: Yes

Reviewer #2: Yes

2. Is the protocol technically sound and planned in a manner that will lead to a meaningful outcome and allow testing the stated hypotheses?

Reviewer #1: Yes

Reviewer #2: Yes

3. Is the methodology feasible and described in sufficient detail to allow the work to be replicable?

Reviewer #1: Yes

Reviewer #2: Yes

4. Have the authors described where all data underlying the findings will be made available when the study is complete?

Reviewer #1: Yes

Reviewer #2: No

5. Is the manuscript presented in an intelligible fashion and written in standard English?

Reviewer #1: Yes

Reviewer #2: Yes

You may also provide optional suggestions and comments to authors that they might find helpful in planning their study.

Reviewer #1: Song and colleagues are planning to perform a systematic review and meta-analysis, here outlining their planned work in an overall well-written and constructed protocol. A PROSPERO version of the protocol is available, and as of 10th of July, it appears as if the work is at the stage of screening. Preferrably, the authors should have waited for at least the initial reviewer feedback before commencing the formal search and screening, but it is, at the same time, fully understandable that they have now, and in practical terms, this is often the reality. The topic is clinically relevant, with a novel angle, and I applaud the authors for submitting a detailed protocol! In general, I see no major issues with this protocol or the methodological approaches outlined; however, there are bits here and there that can benefit from some polishing and revision. I've outlined those below. Please apologize the many points, some of them are relatively trivial and have more the nature of questions.

PROSPERO (just three trivial questions)

- why did you switch the author order (between the PROSPERO protocol and this protocol)?

is any of the authors a methodologist (in meta-research/evidence synthesis)/librarian, or has a person with such competence been consulted throughout the planning of this SR/MA? (e.g., for double-checking the search queries or the like)

- is the "Current review stage" indeed up-to-date? Because as I see, the first three steps have been marked as "started", but neither of the first two have been marked as "completed"...

Title

- Why is the focus/(choice of word) on *mitigating* life-style factors? Usually, from what i see, life-style factors investigated are rather *risk* factors...

Abstract

- From the Background, it's not clear why a SR/MA is needed. Is it because the results from primary studies are conflicting? Is it that it's unclear which *specific* cognitive outcomes the associations are actually robust/consistent in? Something else?

- In Methods, you write "[...] in older adults with CMM"... but what's the control group then? Please clarify/specify this part

- In Methods, you write about two independent reviewers doing screening, but what about data extraction and quality assessment? Those should be performed in pairs as well

Introduction

- Please clarify the difference between cognitive decline and cognitive impairment

- I'd simplify line 61 to "two-fold"/"twice"

- The sentence on lines 66-68 feels unfinished..

- For the part on lines 88-93, please add country and mean age at baseline where it's missing

- On line 97, is it meant "[...] between CMM and cognitive outcomes [...]"?

- Wouldn't the sentence on lines 100-102 be better suited in the very beginning of that paragraph?

- Technically speaking (regarding lines 103-104), Alzheimer's disease is also a type of dementia.. So maybe just "dementia"? why limit yourselves in that regard?

- Again, on line 104, "[...] among older individuals with CMM [...]"... then who is the control group?

- Finally, on line 104, "[...] and 2) Identify [...]"... i think what you mean is to investigate? Also, again, I understand completely the interest in healthy lifestyle factors to adopt, but what about modifiable *risk* factors that can be avoided? it sounds mostly comprehensive when reading your text, but at times quite limiting (what you mean)...

Methods

- Which translation tools will be employed, and why?

- How will you deal with Google Scholar, given that it may easily return enormous amounts of records? Usually, SR/MA take the first 200-300 records or so; how is your plan, and why?

- Including gray literature is very good! but i wonder, is OpenGrey even searchable/indexed? It was more or less de facto defunct last time i checked... How about ProQuest Dissertations and Theses, OAIster, etc...

- I'd strongly recommend searching PubMed instead of Medline, as the former contains ALL of Medline-indexed records PLUS some preprints and other material that could potentially be relevant. Although Medline admittedly offers richer search syntax (e.g., NEAR/x), I'd argue that comprehensiveness outweighs in this case

- Any plans about ensuring non-western inclusion? E.g., i see you plan to use Web of Science, which is great. Please consider also searching KCI and SciELO through WoS (as i recall it doesn't search those by default, just the "core collection"). Also, please consider WHO Global Index Medicus, CABI Direct...

- I'm missing a search query for Google Scholar/OpenGrey

- Why the cut-off of ≥50 years (at baseline, i assume)?

- Hmm, reading the Comparators section, it remains a bit unclear to me, what will *actually* be investigated.. because based on lines 147-149, you are not searching for studies with control groups being without (as many) cardiometabolic diseases... so are you indeed *only* interested in populations where ≥2 CMDs are present?

- Are you sure that case-control studies qualify as longitudinal? Judging by the ROBINS-E paper, it's also not exactly geared for it ("Future work will produce variants of the tool for other epidemiologic study designs (e.g. case-control studies)."; https://pmc.ncbi.nlm.nih.gov/articles/PMC11098530). To strenghten the rigor and clinical utility of this SR/MA, i'd refrain from including case-control studies and keep it at prospective cohort studies, historical cohort data, case-cohort studies, nested case-control studies (i.e., within a cohort), etc..

- Please reconsider excluding letters to the editor. At least from my experience, the trend in some high-impact journals recently is to downgrade many good-quality articles to letters. Some of the authors then opt for putting most of the now "extra" material/data to suppl. material and online repositories (thus, the same data and rigor/detail is actually there, as in a "full-length" article)...

- Will you employ any definition of minimum follow-up length? Would it make sense for some outcomes, given how slowly they develop? Or is this something that you'll simply just comment on, if you see fit?

- I'd place (and also *do*) Data Extraction prior to Assessment of methodological quality

- I assume you might want to revise the estimated deadline on line 197?

- I agree that HR ≈ RR in case the outcome is rare (<15% or so), but i would say it's not quite the case otherwise (please see Table 2 in https://doi.org/10.7326/M16-2607)

- Please clarify (or rather, specify) the meta-analysis models you want to employ. I assume random-effects, although you specifically mention "mixed" modelling multiple times; what's the rationale and what setup are you expecting? Which method of assessment of between-study variance etc? Also, please consider looking into tau(2), as it's more robust than I2 for small sample sizes, if you fear such scenarios in particular

- Why meta-regression for RQ2? Do you really expect such data?

- Subgroup/sensitivity analyses seem fine, but why no sensitivity analysis excluding low-quality studies (for example)?

- Please consider making your R code publicly available, together with collected data (for meta-analysis); a great example/inspiration can be found here: https://journals.plos.org/plosmedicine/article?id=10.1371/journal.pmed.1003731

Discussion

- Is there a reference or a further specified rationale for what is said about IPD MA on lines 256-258? My personal point of view is that the heterogeneity seen in clinical context is so great that MAs in and of themselves are by default quite "rough" and always need to be examined cautiously, and i haven't seen proof of IPD MA making things better per se (in this context)

Reviewer #2: This is a well structured manuscript from abstract to reference. But, the author did not include any tables or diagram to show what was done to back up the analysis. No evidence of the research carried out was involved in this manuscript.

**Do you want your identity to be public for this peer review?** For information about this choice, including consent withdrawal, please see our Privacy Policy

Reviewer #1: **Yes: ** Daniil Lisik

Reviewer #2: No

---

## [Author Response · Author response to Decision Letter 1]

3 Oct 2025

Response to editors and reviewers

Dear editors and reviewers,

We would like to sincerely thank you for your constructive comments and valuable feedback on our manuscript. Your suggestions have greatly helped us to improve the clarity, rigor, and compliance with the journal’s requirements. We have carefully revised the manuscript and addressed each point raised. Below, we provided a detailed, point-by-point response to the comments.

Response to the editor

Journal Requirements:

https://journals.plos.org/plosone/s/file? id=wjVg/PLOSOne_formatting_sample_main_body.pdf and https://journals.plos.org/plosone/s/file? id=ba62/PLOSOne_formatting_sample_title_authors_affiliations.pdf

We checked carefully and ensured that the manuscript meets PLOS ONE style requirements.

[This work was supported by the Trinity College Dublin-China Scholarship Council PhD Scholarship (Grant Number 202406920021). The funders had no role in study design, data collection and analysis, decision to publish, or preparation of the manuscript.].

Please provide an amended statement that declares *all* the funding or sources of support (whether external or internal to your organization) received during this study, as detailed online in our guide for authors at http://journals.plos.org/plosone/s/submit-now.Please also include the statement “There was no additional external funding received for this study.” in your updated Funding Statement.

3. Thank you for stating the following in your manuscript:

[This work was supported by the Trinity College Dublin-China Scholarship Council PhD Scholarship (Grant Number 202406920021). The funders had no role in study design, data collection and analysis, decision to publish, or preparation of the manuscript.]

[JS received the Trinity College Dublin-China Scholarship Council PhD Scholarship (Grant Number 202406920021). The funders had no role in study design, data collection and analysis, decision to publish, or preparation of the manuscript.]

We have removed the funding information from the manuscript and amended the Funding Statement as follows in the cover letter:

“JS received the Trinity College Dublin-China Scholarship Council PhD Scholarship (Grant Number 202406920021). The funders had no role in study design, data collection and analysis, decision to publish, or preparation of the manuscript. Proof of an independent peer review by the funder is not applicable in this case, as the funding was awarded as a general PhD scholarship. There was no additional external funding received for this study.”

We appreciate your assistance in updating the online submission form on our behalf.

4. When completing the data availability statement of the submission form, you indicated that you will make your data available on acceptance. We strongly recommend all authors decide on a data sharing plan before acceptance, as the process can be lengthy and hold up publication timelines. Please note that, though access restrictions are acceptable now, your entire data will need to be made freely accessible if your manuscript is accepted for publication. This policy applies to all data except where public deposition would breach compliance with the protocol approved by your research ethics board. If you are unable to adhere to our open data policy, please kindly revise your statement to explain your reasoning, and we will seek the editor's input on an exemption. Please be assured that, once you have provided your new statement, the assessment of your exemption will not hold up the peer review process.

Thank you for your guidance regarding the data availability statement. We agree with the journal’s open data policy and will ensure that all data is made freely accessible upon acceptance for publication.

We have ensured that the ethics statement now only appears under the Method section “Ethics and dissemination” of the manuscript.

Response to reviewers

Reviewer #1

Song and colleagues are planning to perform a systematic review and meta-analysis, here outlining their planned work in an overall well-written and constructed protocol. A PROSPERO version of the protocol is available, and as of the 10th of July, it appears as if the work is at the stage of screening. Preferably, the authors should have waited for at least the initial reviewer feedback before commencing the formal search and screening, but it is, at the same time, fully understandable that they have now, and in practical terms, this is often the reality. The topic is clinically relevant, with a novel angle, and I applaud the authors for submitting a detailed protocol! In general, I see no major issues with this protocol or the methodological approaches outlined; however, there are bits here and there that can benefit from some polishing and revision. I've outlined those below. Please apologize for the many points, some of them are relatively trivial and have more the nature of questions.

Thank you for your detailed and constructive comments. We sincerely appreciate your recognition of the clinical relevance and novelty of our topic.

Regarding the timing of the formal search and screening, we acknowledge your point. However, given the typically long waiting time for initial review feedback, we deemed it more practical to initiate the formal search and screening in advance. These processes, however, were conducted under the guidance of a librarian, and the research team received appropriate training to ensure methodological rigor.

We are grateful for your careful review and for highlighting areas that could benefit from further refinement. We carefully addressed all comments and revised the manuscript accordingly.

PROSPERO (just three trivial questions)

- why did you switch the author order (between the PROSPERO protocol and this protocol)? is any of the authors a methodologist (in meta-research/evidence synthesis)/librarian, or has a person with such competence been consulted throughout the planning of this SR/MA? (e.g., for double-checking the search queries or the like)

Sorry for the confusion this caused. Yes, there is a librarian involved in our project. The initial PROSPERO entry did not strictly follow the intended author order, and we amended the order in the manuscript to ensure consistency.

- Is the "Current review stage" indeed up-to-date? Because, as I see, the first three steps have been marked as "started", but neither of the first two has been marked as "completed"...

Thanks for pointing this out. However, the “current review stage” in PROSPERO is not always updated in real time. We have now updated the PROSPERO entry accordingly to reflect the current progress.

Title

- Why is the focus/(choice of word) on *mitigating* life-style factors? Usually, from what I see, lifestyle factors investigated are rather *risk* factors...

We understand the potential confusion. While many studies indeed investigate lifestyles as risk factors, our review specifically focuses on their mitigating role, examining how they act as moderators or mediators that lessen the adverse effects of CMM on cognitive outcomes. We therefore chose the term *mitigating* to better reflect the scope and purpose of our review.

However, your comments prompted us to critically reconsider, and we now realise that the revised wording aligns better with the aim of this review. The change of title is consistent with our research question and previous search strategies.

Line 1-2:

Title changed:

“The role of lifestyle in mitigating cognitive decline in older adults with cardiometabolic multimorbidity (CMM): A protocol of systematic review and meta-analysis”

Abstract

- From the Background, it's not clear why a SR/MA is needed. Is it because the results from primary studies are conflicting? Is it that it's unclear which *specific* cognitive outcomes the associations are actually robust/consistent in? Something else?

Thank you for this comment. We have revised the background to clarify the rationale for conducting this SR/MA.

Line 22-27:

“Cardiometabolic multimorbidity (CMM), the coexistence of two or more cardiometabolic diseases (CMDs), is increasingly recognized as a key risk factor for cognitive decline and dementia in older adults. Healthy lifestyle behaviors may mitigate these effects, but this remains poorly understood. Given the rising prevalence of CMM worldwide and the urgent need for effective dementia prevention strategies, a systematic review and meta-analysis are warranted to investigate whether and to what extent a healthier lifestyle can mitigate this risk.”

- In Methods, you write "[...] in older adults with CMM"... but what's the control group then? Please clarify/specify this part

Thank you for your comment. We phrased it this way to align with the main research terms. Our focus is on the primary research question, which compares individuals with less healthy lifestyles to those with healthier lifestyles, within the context of CMM.

No changes made.

- In Methods, you write about two independent reviewers doing screening, but what about data extraction and quality assessment? Those should be performed in pairs as well

Yes, data extraction and quality assessment will also be conducted by two reviewers independently. We explained this in the Method section; however, it was not included in the abstract due to word limits.

Line 32-33:

“Retrieved records will be managed in Covidence, with two independent reviewers conducting all review processes.”

Introduction

- Please clarify the difference between cognitive decline and cognitive impairment

Thank you for your comments.

Cognitive decline refers to a gradual deterioration in cognitive abilities over time, which may occur in normal aging or precede more serious deficits. Cognitive impairment denotes a measurable deficit at a single point in time, ranging from mild cognitive impairment (MCI) to dementia, and can constitute a categorial clinical diagnosis. Cognitive impairment can be defined as reaching a prespecified criterion of cognitive decline.

We have revised the text and added references to clarify this point.

Line 50-51:

“People with CMDs face an increased risk of developing cognitive decline, cognitive impairment with no dementia (CIND) [2], and ultimately dementia [3–5].”

[2]. Ailshire JA, Farina MP, Jackson H, Walsemann KM. The role of educational attainment and quality in U.S. regional variation in prevalence of dementia and CIND. PLOS ONE. 2025;20: e0332410. doi:10.1371/journal.pone.0332410

[5]. Vishwanath S, Qaderi V, Steves CJ, Reid CM, Hopper I, Ryan J. Cognitive decline and risk of dementia in individuals with heart failure: A systematic review and meta-analysis. J Card Fail. 2022;28: 1337–1348. doi:10.1016/j.cardfail.2021.12.014

- I'd simplify line 61 to "two-fold"/"twice"

Thank you for your advice. We aimed to present the exact evidence by reporting the numerical hazard ratios. However, we have adopted your suggestion and revised the expression to “two-fold” for simplicity.

Line 57-58:

“…have nearly a two-fold risk of developing dementia compared to those without multimorbidity [11].”

- The sentence on lines 66-68 feels unfinished..

You are right. We have revised and restructured the sentence.

Line 68-72:

“Many studies have investigated cognitive health using composite cardiovascular health scores, including indicators of both CMDs and lifestyle factors such as exercise, diet, and sleep [15,16]. While composite scores are useful for predicting cognitive outcomes in the general population, they may offer limited insight for tailoring interventions to individuals with CMM.”

- For the part on lines 88-93, please add country and mean age at baseline where it's missing

Thank you for your advice. We have revised the text and added the missing information regarding country and mean age to ensure clearer comparisons.

Line 60-64:

“Similarly, a 12-year follow-up study of Swedish older adults aged 60 and older found a slightly lower HR of 1.86 (95% CI 1.17 to 2.97) [17]. While data from the Health and Retirement Study, with a mean follow-up of 14.6 years, showed a higher HR of 3.27 (95% CI 2.06 to 5.21) in American participants with a mean age of 59.4 years [18].”

- On line 97, is it meant "[...] between CMM and cognitive outcomes [...]"?

Yes. We have changed accordingly.

Line 94:

“…CMM on cognitive health”

- Wouldn't the sentence on lines 100-102 be better suited in the very beginning of that paragraph?

You are correct. We have made the changes accordingly, moving the sentence “To date, no systematic review has synthesized the data regarding the effect of CMM on cognitive outcomes in older adults or the potential mitigating role of lifestyle factors in this relationship” to the beginning of the paragraph. (Line 89-90)

- Technically speaking (regarding lines 103-104), Alzheimer's disease is also a type of dementia.. So maybe just "dementia"? Why limit yourselves in that regard?

You are right. We have deleted “Alzheimer’s disease”.

Line 95:

“…including cognitive decline, CIND, and dementia”

- Again, on line 104, "[...] among older individuals with CMM [...]"... then who is the control group?

We hope our previous response to the same question provides clarification on this point.

- Finally, on line 104, "[...] and 2) Identify [...]"... I think what you mean is to investigate? Also, again, I understand completely the interest in healthy lifestyle factors to adopt, but what about modifiable *risk* factors that can be avoided? it sounds mostly comprehensive when reading your text, but at times quite limiting (what you mean)...

Thank you for this comment. We believe the confusion arose from the use of the word “identify”. You are correct that “investigate” is a more accurate term than “identify” in this context.

Line 93: substitute “identify” with “investigate”

Methods

- Which translation tools will be employed, and why?

Google Translate will be used when necessary, and we have added the relevant information with references to clarify this in the text.

Line 116-118:

“Studies published in all languages will be included to ensure methodological rigor, with Google Translate employed when necessary as a viable and accurate tool for translating in systematic reviews [23,24].”

[23]. Davidson SL, Lee J, Emmence L, Bickerstaff E, Rayers G, Davidson E, et al. Systematic review and meta-analysis of the prevalence of frailty and pre-frailty amongst older hospital inpatients in

---

## [Decision Letter · Decision Letter 1]

16 Oct 2025

Dear Dr. SONG,

Thank you for submitting your manuscript to PLOS ONE. After careful consideration, we feel that it has merit but does not fully meet PLOS ONE’s publication criteria as it currently stands. Therefore, we invite you to submit a revised version of the manuscript that addresses the points raised during the review process.

We look forward to receiving your revised manuscript.

Kind regards,

Tai Dinh

Academic Editor

PLOS ONE

Journal Requirements:

Additional Editor Comments:

Please further revise the manuscript to address all comments raised by the reviewer.

Reviewers' comments:

Reviewer's Responses to Questions

**Comments to the Author**

1. Does the manuscript provide a valid rationale for the proposed study, with clearly identified and justified research questions?

Reviewer #1: Yes

Reviewer #2: Yes

2. Is the protocol technically sound and planned in a manner that will lead to a meaningful outcome and allow testing the stated hypotheses?

Reviewer #1: Yes

Reviewer #2: Yes

3. Is the methodology feasible and described in sufficient detail to allow the work to be replicable?

Reviewer #1: Yes

Reviewer #2: Yes

4. Have the authors described where all data underlying the findings will be made available when the study is complete?

Reviewer #1: Yes

Reviewer #2: Yes

5. Is the manuscript presented in an intelligible fashion and written in standard English?

Reviewer #1: Yes

Reviewer #2: Yes

You may also provide optional suggestions and comments to authors that they might find helpful in planning their study.

Reviewer #1: The authors have provided thorough and professional responses and appropriate revisions. I am happy.

Reviewer #2: Thank you for your response. But, you only attached a checklist for the analysis and not the analysis itself, which do not suffice in my opinion, I understand your point about availability at the moment. You also need to know that, the analysis is important for your research paper.

**Do you want your identity to be public for this peer review?** For information about this choice, including consent withdrawal, please see our Privacy Policy

Reviewer #1: **Yes: ** Daniil Lisik

Reviewer #2: No

---

## [Author Response · Author response to Decision Letter 2]

4 Nov 2025

Reviewer #1: The authors have provided thorough and professional responses and appropriate revisions. I am happy.

Thank you for your positive feedback and for acknowledging our efforts. We sincerely appreciate your advice and guidance.

Reviewer #2: Thank you for your response. But, you only attached a checklist for the analysis and not the analysis itself, which do not suffice in my opinion, I understand your point about availability at the moment. You also need to know that, the analysis is important for your research paper.

Thank you for your comment.

We have carefully considered your previous feedback: “This is a well-structured manuscript from abstract to reference. But the author did not include any tables or diagrams to show what was done to back up the analysis. No evidence of the research carried out was involved in this manuscript.”

Given that the inclusion and exclusion criteria were clearly defined in the “Eligibility Criteria” section based on Population, Exposure, Comparator, and Outcomes (PECOS), we have additionally provided a partially completed PRISMA diagram (Fig. 1) alongside the R code to enhance clarity of the analyses. The R code has been validated using a simulation dataset to ensure reproducibility and is also accessible on GitHub (https://github.com/songj4-byte/HL_CMM_Cog_SRMA2025). Writing the R code helped clarify the analysis steps and some details. We also removed certain content to improve readability and understanding.

Changes made:

• Line 161: Added “(Fig. 1)”

• Line 196: Removed the sentence: “The primary research question will be addressed using meta-regression, modelling lifestyle indicators as study-level moderators of the association between CMM and cognitive outcomes.”

---

## [Editor Report · Decision Letter 2]

30 Nov 2025

The role of lifestyle in mitigating cognitive decline in older adults with cardiometabolic multimorbidity (CMM): A protocol of systematic review and meta-analysis

PONE-D-25-25646R2

Dear Dr. SONG,

We’re pleased to inform you that your manuscript has been judged scientifically suitable for publication and will be formally accepted for publication once it meets all outstanding technical requirements.

Kind regards,

Tai Dinh

Academic Editor

PLOS ONE

Additional Editor Comments (optional):

The authors have revised the paper according to the reviewers’ suggestions, and I have therefore decided to accept it.
---

## [Editor Report · Acceptance letter]

PONE-D-25-25646R2

PLOS One

Dear Dr. SONG,

I'm pleased to inform you that your manuscript has been deemed suitable for publication in PLOS One. Congratulations! Your manuscript is now being handed over to our production team.

Kind regards,

on behalf of

Dr. Tai Dinh

Academic Editor

PLOS One